# Dealing with aflatoxin $B_1$ dihydrodiol acute effects: Impact of aflatoxin $B_1$-aldehyde reductase enzyme activity in poultry species tolerant to $AFB_1$ toxic effects

Hansen Murcia ◉°*, Gonzalo J. Diaz°

Laboratorio de Toxicología, Facultad de Medicina Veterinaria y Zootecnia, Universidad Nacional de Colombia, Bogotá D.C., Colombia

° These authors contributed equally to this work.
* hwmurciag@unal.edu.co

**Data Availability Statement:** All relevant data are within the manuscript and its Supporting Information files.

## Abstract

Aflatoxin $B_1$ aldehyde reductase (AFAR) enzyme activity has been associated to a higher resistance to the aflatoxin $B_1$ ($AFB_1$) toxicity in ethoxyquin-fed rats. However, no studies about AFAR activity and its relationship with tolerance to $AFB_1$ have been conducted in poultry. To determine the role of AFAR in poultry tolerance, the hepatic *in vitro* enzymatic activity of AFAR was investigated in liver cytosol from four commercial poultry species (chicken, quail, turkey and duck). Specifically, the kinetic parameters $V_{max}$, $K_m$ and intrinsic clearance ($CL_{int}$) were determined for $AFB_1$ dialdehyde reductase ($AFB_1$-monoalcohol production) and $AFB_1$ monoalcohol reductase ($AFB_1$-dialcohol production). In all cases, $AFB_1$ monoalcohol reductase activity saturated at the highest aflatoxin $B_1$ dialdehyde concentration tested (66.4 $\mu$M), whereas $AFB_1$ dialdehyde reductase did not. Both activities were highly and significantly correlated and therefore are most likely catalyzed by the same AFAR enzyme. However, it appears that production of the $AFB_1$ monoalcohol is favored over the $AFB_1$ dialcohol. The production of alcohols from aflatoxin dialdehyde showed the highest enzymatic efficiency (highest $CL_{int}$ value) in chickens, a species resistant to $AFB_1$; however, it was also high in the turkey, a species with intermediate sensitivity; further, $CL_{int}$ values were lowest in another tolerant species (quail) and in the most sensitive poultry species (the duck). These results suggest that AFAR activity is related to resistance to the acute toxic effects of $AFB_1$ only in chickens and ducks. Genetic selection of ducks for high AFAR activity could be a means to control aflatoxin sensitivity in this poultry species.

## Introduction

Aflatoxin $B_1$ ($AFB_1$) is a secondary metabolite produced by some strains of *Aspergillus* fungi, including *Aspergillus flavus*, *A. parasiticus*, *A. nomius* and *A. pseudonomius*. Hepatic biotransformation of $AFB_1$ by cytochrome P450 (CYP) enzymes produces aflatoxin $B_1$-8,9-epoxide

**Funding:** HM, Convocatoria 647, Departamento Administrativo de Ciencia, Tecnología e Innovación, COLCIENCIAS, https://www.colciencias.gov.co/convocatorias/2014/doctorados-nacionales-2014, The funders had no role in study design, data collection and analysis, decision to publish, or preparation of the manuscript. GD, Departamento de Salud Animal, Facultad de Medicina Veterinaria y Zootecnia, Universidad Nacional de Colombia, Bogotá D.C., Colombia, http://medicinaveterinariaydezootecnia.bogota.unal.edu.co/la-facultad/departamentos/, The funders had no role in study design, data collection and analysis, decision to publish, or preparation of the manuscript.

**Competing interests:** The authors have declared that no competing interests exist.

(AFBO) with two possible stereoisomers: aflatoxin $B_1$-8,9-*exo*-epoxide and aflatoxin $B_1$-8,9-*endo*-epoxide; only aflatoxin $B_1$-8,9-*exo*-epoxide can react with DNA, producing adducts at position 7 of guanine leading to carcinogenesis [1, 2]. Once formed, AFBO is highly electrophilic and quickly reacts with water ($t_{1/2}$ = <1 second) forming $AFB_1$-8,9-dihydrodiol ($AFB_1$-dhd) [3–5]. $AFB_1$-dhd has a pH dependent equilibrium with another species known as $AFB_1$ dialdehyde, which can produce Schiff bases with lysine [6], affecting protein synthesis and causing cytotoxic effects [7]. $AFB_1$ dialdehyde can be reduced by aflatoxin $B_1$-aldehyde reductase (AFAR, EC 1.1.1.2; Fig 1), a cytosolic enzyme of the aldo-keto reductase superfamily, that was first described in liver extracts from ethoxyquin-fed rats [8]. A DNA sequence associated with AFAR enzyme activity [9] was later identified as the inducible isoform of aflatoxin $B_1$-aldehyde reductase 1 (*AFAR1*, currently known as *AKR7A1*), which strongly interacts with chemopreventive agents such as ethoxyquin [10]. After the discovery of the AKR7A1 enzyme, a second encoding region was found corresponding to the enzyme AKR7A3 (formerly known as AFAR29) [11, 12]. Concerning inducible gene expression, it has been observed that both AFAR and glutathione sulfotransferases (GSTs) are inducible by compounds like ethoxyquin, making it difficult to discriminate the relevance of these two enzymatic activities on $AFB_1$ toxicity [13, 14].

Information on AFAR enzyme activity in poultry is scarce; however, sequences corresponding to AFAR enzymes have been reported for poultry species. In the National Center for Biotechnology Information [15] there is a chicken DNA sequence which corresponds to the

**Fig 1. Bioactivation of aflatoxin B₁ into aflatoxin B₁ 8,9-epoxide through cytochrome P450 enzymes (CYPs).** Spontaneous hydrolysis of the epoxide or the enzymatic activity of epoxide hydrolase (EPHX), produce aflatoxin B₁ 8,9-dihydrodiol which in a pH dependent manner equilibrates with aflatoxin B₁ dialdehyde. Enzymatic reduction of aflatoxin B₁ dialdehyde into aflatoxin B₁ C6-monoalcohol and aflatoxin B₁ C8-monoalcohol is carried out by aflatoxin B₁ aldehyde reductase (AFAR), which in turn can also reduce these monoalcohols into aflatoxin B₁ dialcohol.

*AKR7A2* gene. Furthermore, in the Kegg Pathways database [16–18] the AKR7A2 enzyme is associated with aflatoxin B$_1$ dialdehyde reduction to AFB$_1$-C6-monoalcohol phenolate and AFB$_1$-C8-monoalcohol phenolate; these two monoalcohols can be further reduced to AFB$_1$-dialcohol phenolate. Other poultry DNA sequences with functional annotations found in the NCBI [15] include a sequence in turkeys that corresponds to aflatoxin B$_1$ aldehyde reductase member 2-like, and sequences in ducks and quail that correspond to *AKR7A2* aldo-keto reductase family 7, member A2.

Because it has been suggested that AFAR activity is related to a higher resistance to AFB$_1$, especially through ameliorating the acute effects caused by aflatoxin dialdehyde [8], the present study was conducted to investigate the enzyme kinetic parameters of aflatoxin B$_1$-monoalcohol and aflatoxin B$_1$-dialcohol production, and to relate them with the known sensitivity to AFB$_1$ in chickens, turkeys, ducks and quail.

## Materials and methods

### Reagents

Glucose 6-phosphate sodium salt, glucose 6-phosphate dehydrogenase, nicotinamide dinucleotide phosphate (NADP$^+$), ethylenediaminetetraacetic acid (EDTA), bicinchoninic acid solution (sodium carbonate, sodium tartrate, sodium bicarbonate and sodium hydroxide 0.1 N pH 11.25), copper sulphate pentahydrate, formic acid, sucrose, bovine serum albumin, sodium borohydride, *m*-chloroperbenzoic acid, ethanol (spectrophotometric grade), isopropyl alcohol, 2-(cyclohexylamino) ethane sulfonic acid (CHES), aflatoxin B$_{2a}$ and *N,N*-dimethylformamide were from Sigma-Aldrich (St. Louis, MO, USA). Aflatoxin B$_1$ was from Fermentek Ltd. (Jerusalem, Israel). Sodium phosphate monobasic monohydrate, sodium phosphate dibasic anhydrous and sodium chloride were from Merck (Darmstadt, Germany). Methanol, acetonitrile and water were all HPLC grade.

### AFB$_1$-dhd synthesis and purification

AFB$_1$-dhd was produced based on the method of Fringuelli [19] with some modifications. To a 2 mL of a water:acetonitrile mix (1:1, v/v), 5 mg of AFB$_1$ and 5.38 mg of *m*-chloroperbenzoic acid ≤ 70% were added and mixed. The mix was stirred at room temperature for 30 minutes, after which the AFB$_1$-dhd formed was purified by using a $\mu$Bondapack C18 125 Å, 10 $\mu$m, 7.8 x 300 mm preparative column (Waters Corporation, Milford, MA, USA) kept at 50˚C. The chromatograph was an Agilent Technologies InfinityLab LC system (Agilent, Santa Clara, CA, USA) equipped with a G1314B 1260 VWD VL variable wavelength UV/Vis detector, a G1316A 1260 TCC thermostated column compartment, a G1329B 1260 ALS standard autosampler, and a G1311C 1260 Quaternary Pump VL, all modules controlled by "LC Openlab CDS ChemStation Edition" software. The mobile phase was a linear gradient of solvents A (water 0.1% formic acid) and B (isopropil alcohol 20% in acetonitrile, 0.1% formic acid) as follows: 0 min: 18% B, 10 min: 18% B, 13 min: 100% B, 15 min: 100% B, 15.01 min: 18% B, 17 min: 18% B. The flow rate was 2.5 mL/min and the UV detector was set at 365 nm. Aliquots of 50 $\mu$L of the synthesis solution were injected until all the volume was run into the HPLC system. The AFB$_1$-dhd-containing fractions were collected, taken to dryness in a rotary evaporator (Hei-Vap Advantage, Heidolph Instruments GmbH & CO, Schwabach, Germany) and resuspended in ultrapure water. The purified AFB$_1$-dhd was quantitated using an external standard of AFB$_{2a}$, since these two compounds share identical spectral properties [20].

## AFB$_1$ monoalcohol and dialcohol synthesis and purification

The synthesis of AFB$_1$ monoalcohol and AFB$_1$ dialcohol was made according to the method of Guengerich [21]. To a 1 mL of a 63 $\mu$M solution of AFB$_1$-dhd in water acetonitrile 1:1 (v/v) (adjusted to pH 10 with buffer CHES 250 mM), 60 $\mu$L of an 8.9 mM solution of NaBH$_4$ in *N, N*-dimethylformamide was added and let stir for 30 minutes. After this, 10 $\mu$L of formic acid was added to neutralize the mixture. Purification of the synthesis products was made by pre-parative HPLC on a Phenomenex Prodigy LC Column C18 ODS-3V 100 Å, 250 x 4.6 mm 5 $\mu$m (Phenomenex, Torrance CA. USA) kept at 40˚C. The mobile phase was a linear gradient of solvents A (water 0.1% formic acid) and B (acetonitrile 0.1% formic acid) as follows: 0 min: 15% B, 5 min: 15% B, 15 min: 40% B, 15.01 min: 100% B, 17 min: 100% B, 17.01min: 15% B, 27 min: 15% B. The flow rate was set at 0.6 mL/min and the UV detector was set at 365 nm. Aliquots of 10 $\mu$L were injected until the whole synthesis volume was run in the HPLC system. The fractions containing the compounds were collected, taken to dryness in a rotary evaporator (Hei-Vap Advantage, Heidolph Instruments GmbH & CO, Schwabach, Germany) and suspended in ethanol for UV quantitation. The concentrations of the AFB$_1$ monoalcohol and AFB$_1$ dialcohol were estimated by using the AFB$_1$ extinction coefficient in ethanol ($\epsilon$ = 21800 $M^{-1}$ $cm^{-1}$; [22]). To confirm their identities, the monoisotopic protonated masses of both compounds were determined by HPLC-MS on a Sciex 3200 QTrap mass spectrometer (Applied Biosystems, Toronto, Canada) using a thermospray ionization probe in positive mode and the following settings: probe voltage = 4,800 V, declustering potential = 140 V, entrance potential = 10 V, curtain gas value: 30, collision energy = 81 V and collision cell exit potential = 5 V. Since the molecular masses of both AFB$_1$ C6-monoalcohol and AFB$_1$ C8-monoalcohol are the same, and only one chromatographic peak was found, the enzyme kinetics analyses were done for both analytes under the term AFB$_1$ monoalcohol.

## Microsomal and cytosolic fraction processing

Liver fractions were obtained from 12 healthy birds (6 males and 6 females) from each of the following species and age: seven-week old Ross and Rhode Island Red chickens (*Gallus gallus ssp. domesticus*), eight-week old Nicholas turkeys (*Meleagris gallopavo*), eight-week old Japanese quails (*Coturnix coturnix japonica*) and nine-week old meat-type Pekin ducks (*Anas platyrhynchos ssp. domesticus*). No additives or medication were added to the diets provided to the birds. The diets were formulated with the same ingredients (corn, extruded full-fat soybeans, soybean meal, vegetable oil, calcium phosphate, calcium carbonate, sodium chloride, lysine, methionine, tryptophan, choline, vitamin and mineral premix) formulated to reach or exceed the nutrient requirements of each poultry species studied. Poultry were obtained from local commercial poultry suppliers and at the moment of sacrifice no noticeable clinical signs were observed. The birds were sacrificed by cervical dislocation, and their livers extracted immediately, washed with cold PBS buffer (50 mM phosphates, pH 7.4, NaCl 150 mM), cut into small pieces and stored at −70˚C until processing. The experiment was conducted following the welfare guidelines of the Poultry Research Facility and was approved by the Bioethics Committee, Facultad de Medicina Veterinaria y Zootecnia, Universidad Nacional de Colombia, Bogotá D.C., Colombia (approval document CB-FMVZ-UN-033-18). Frozen liver samples were allowed to thaw, and 2.5 g were minced and homogenized for 1 minute with a tissue homogenizer (Cat X120, Cat Scientific Inc., Paso Robles, CA, USA) after adding 10 mL of extraction buffer (phosphates 50 mM pH 7.4, EDTA 1 mM, sucrose 250 mM). The homogenates were then centrifuged at 12,000 × g for 30 minutes at 4˚C (IEC CL31R Multispeed Centrifuge, Thermo Scientific, Waltham, MA, USA). The resulting supernatants (approximately 10 mL) were transferred into ultracentrifuge tubes kept at 4˚C and centrifuged for 90 minutes

at 100,000 × g (Sorval WX Ultra 100 Centrifuge, Thermo Scientific, Waltham, MA, USA). An aliquot of each of the ultracentrifuged supernatants (corresponding to the cytosolic fraction) was taken to determine its protein content by the bicinchoninic acid protein quantification method according to Redinbaugh and Turley [23]. The remaining supernatant was fractioned into microcentrifuge tubes and stored at −70˚C until used for the enzyme kinetic studies. No further enzyme purification was carried out and the incubations were carried out with the cytosolic fractions obtained as previously described.

## Aflatoxin B$_1$ monoalcohol and AFB$_1$ dialcohol enzyme kinetics

To determine the enzyme kinetics of AFB$_1$ monoalcohol and AFB$_1$ dialcohol production (reduction of AFB$_1$ dialdehyde by AFAR: AFB$_1$ dialdehyde + NADH + H$^+$ → AFB$_1$ monoalcohol + NAD$^+$; reduction of AFB$_1$ monoalcohol by AFAR: AFB$_1$ monoalcohol + NADH + H$^+$ → AFB$_1$ dialcohol + NAD$^+$), the method proposed by Judah *et al*. [8] was used with some modifications. AFB$_1$ dialdehyde was obtained by adjusting to pH 10 (with buffer CHES 25 mM) the chemically synthetized AFB$_1$-dhd [4]. For AFB$_1$ dialdehyde and AFB$_1$ monoalcohol reductase enzyme kinetics, a discontinuous direct assay was carried out in 1.5 mL microcentrifuge tubes kept at 39˚C (the normal body temperature for the age of the birds used) containing 5 mM glucose 6-phosphate, 0.5 IU of glucose 6-phosphate dehydrogenase, 0.5 mM NADP$^+$ and 30 µg of cytosolic protein for chicken breeds or turkey, 50 µg for duck and 70 µg for quail. All volumes were completed with incubation buffer (phosphates 50 mM pH 7.4). After 3 minutes of preincubation, 4 µL of AFB$_1$-dhd in buffer CHES 25 mM pH 10 (AFB$_1$ dialdehyde form) at concentrations ranging from 3.38 to 66.4 µM was added and the reaction stopped after 90 seconds with 250 µL of ice-cold acetonitrile. The stopped incubations were centrifuged at 15,000 × g for 10 minutes and 5 µL were analyzed by HPLC. The amount of AFB$_1$ monoalcohol and AFB$_1$ dialcohol found in each incubation was quantitated in a Shimadzu Prominence system (Shimadzu Scientific Instruments, Columbia, MD, USA) equipped with a DGU-20A3R degassing unit, two LC-20AD pumps, a SIL-20ACHT autosampler with cooling system, a CTO-20A column oven, an RF-20AXS fluorescence detector, and a CBM-20A bus module, all controlled by "LC Solutions" software. The chromatography was carried out on an Alltech Alltima HP C18, 150 mm × 3.0 mm (Alltech Associates Inc., Deerfield, IL, USA) kept at 40˚C. The mobile phase was a linear gradient of solvents A (water − 0.1% formic acid) and B (acetonitrile − 0.1% formic acid), as follows: 0 min: 5% B, 1 min: 5% B, 15 min: 15% B, 15.01 min: 5% B, 20 min: 5% B. The flow rate was 0.6 mL/min and the fluorescence detector was set at excitation and emission wavelengths of 360 nm and 440 nm, respectively. The in-vial concentrations of AFB$_1$ monoalcohol and AFB$_1$ dialcohol were quantitated by using the standards of AFB$_1$ monoalcohol and AFB$_1$ dialcohol chemically synthetized as previously described.

## Statistical analysis

The enzymatic parameters K$_m$ and V$_{max}$ were determined by non-linear regression using the Marquardt method adjusting the data to the Michaelis-Menten enzyme kinetics using the equation: $v = V_{max}[S]/K_m + [S]$, where v is the enzyme reaction velocity, [S] represents substrate concentration, V$_{max}$ represents maximal velocity and K$_m$ represents the Michaelis-Menten constant. Intrinsic clearance (CL$_{int}$—mL/mg protein/minute) was calculated as the ratio V$_{max}$/K$_m$. The calculated CL$_{int}$ only applies for the selected enzymatic activity and not for the hepatic clearance, since AFAR enzyme was not purified from liver extracts. In all cases the kinetic parameters are "apparent" because hepatic extracts and not purified enzymes were used. Inter-species differences in enzymatic kinetic parameters were determined by using the Kruskal-Wallis test, while nonparametric multiple comparisons were made by using the

Dwass-Steel-Critchlow-Fligner method, with a significance level of 5% (p <0.05). Correlations were estimated by using the Spearman's rank-order correlation coefficient. All analyses were performed using the Statistical Analysis System software [24].

## Results

The molecular mass of the chemically-synthetized AFB$_1$ monoalcohol was confirmed by mass spectrometry, since the peak eluting at $t_R$ = 13.92 minutes (Fig 2A) had the expected protonated monoisotopic mass of the compound (349.2 Da; Fig 2B). Similarly, the molecular mass of the AFB$_1$ dialcohol was also confirmed, given that the mass of the peak eluting at $t_R$ = 12.29 minutes (Fig 3A) corresponded to the expected protonated monoisotopic mass (351.0 Da; Fig 3B).

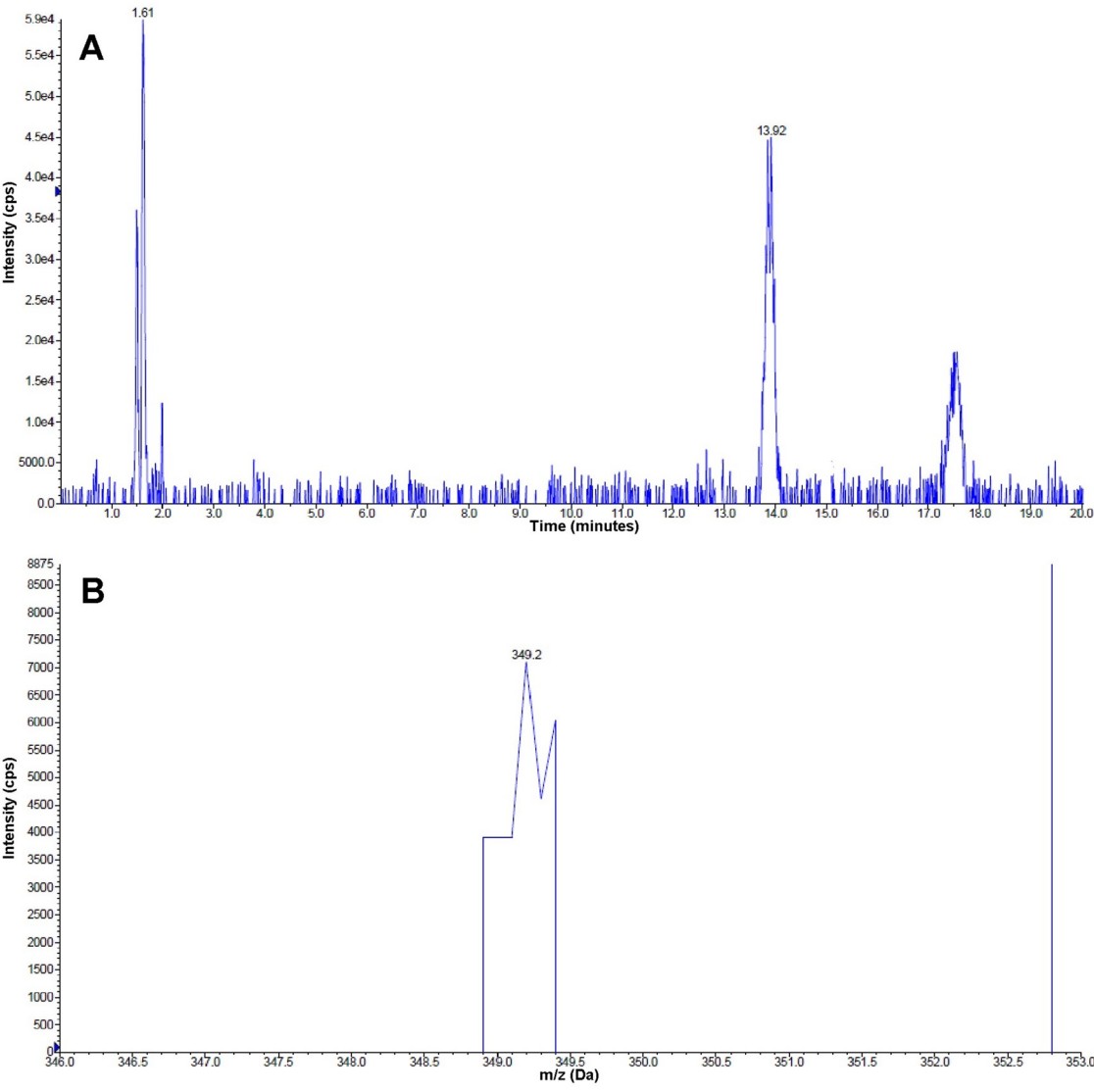

**Fig 2. Identification of AFB$_1$ monoalcohol by HPLC-MS.** (A) Chromatogram of the purified AFB$_1$ monoalcohol product obtained from the reduction of AFB$_1$ dialdehyde with NaBH$_4$. The peak at $t_R$ = 13.92 shows the putative AFB$_1$ monoalcohol product. (B) Protonated monoisotopic mass found in the 13.92 min peak, corresponding to a value of 349.2 Da.

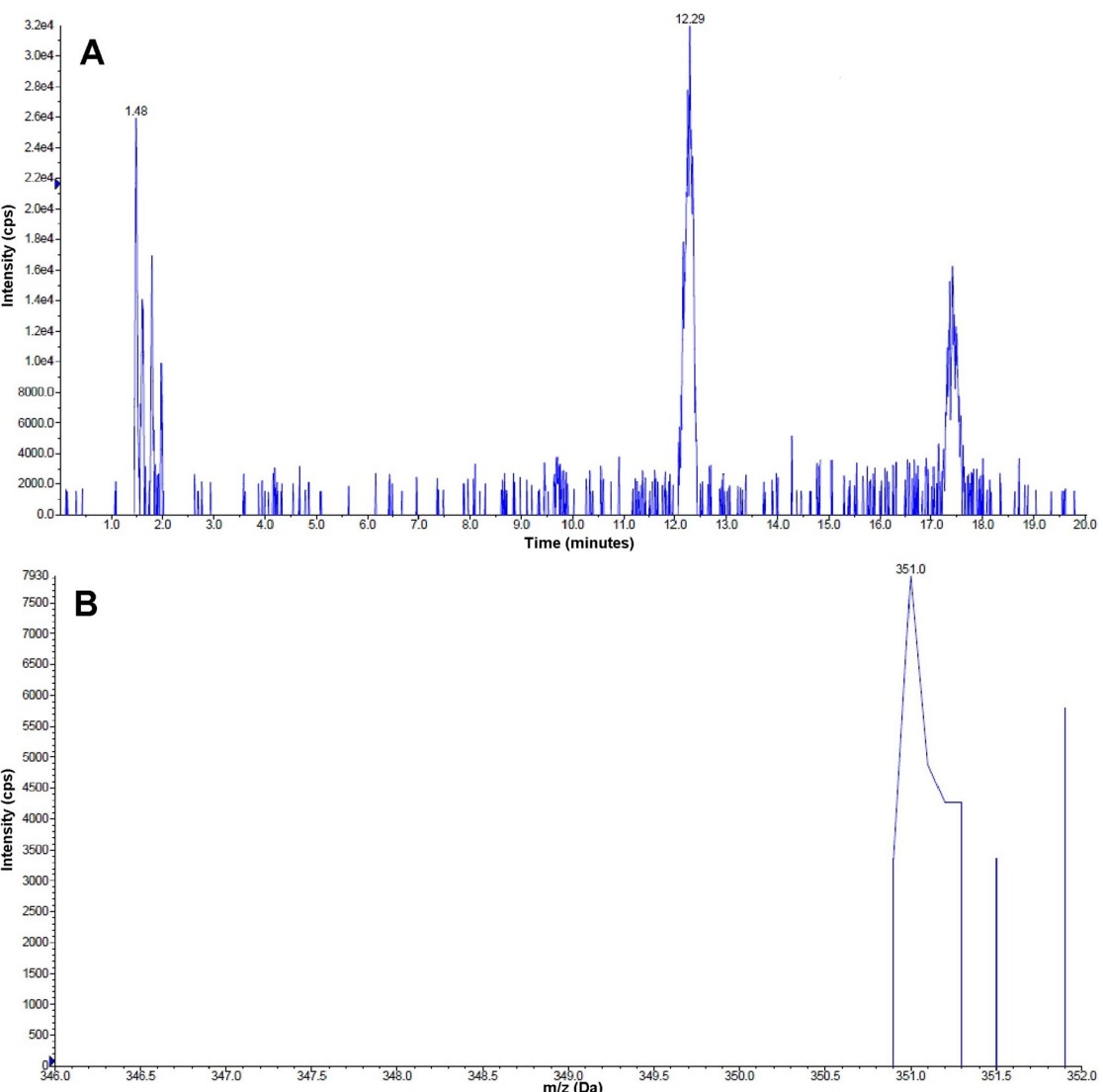

**Fig 3. Identification of AFB$_1$ dialcohol by HPLC-MS.** (A) Chromatogram of the purified AFB$_1$ dialcohol product obtained from the reduction of AFB$_1$ dialdehyde with NaBH$_4$. The peak at $t_R$ = 12.29 shows the putative AFB$_1$ dialcohol product. (B) Protonated monoisotopic mass found in the 12.29 min peak, corresponding to a value of 351.0 Da.

The enzyme kinetics of AFB$_1$-monoalcohol production (AFB$_1$-dialdehyde reductase activity) is shown in Fig 4. The AFAR enzyme activity does not seem to saturate in any of the species studied, even at the highest AFB$_1$-dialdehyde concentration tested (66.4 $\mu$M; Fig 4A). Further, large differences in biotransformation rates were observed, with Rhode Island Red chickens and turkey showing the highest rates, quail and duck the lowest, and the Ross chickens showing an intermediate rate. In regard to the V$_{max}$ kinetic parameter, the Rhode Island Red chickens showed a significantly higher value (10.1 ± 5.52 nmol AFB$_1$ monoalcohol/mg protein/minute), which was about 5 times higher than the values obtained with other poultry (p <0.0001; Fig 4B). No differences in V$_{max}$ were found among the Ross chickens, turkey, quail and duck; however, differences for this parameter were found between sexes for the Ross chickens (1.21 ± 0.38 and 3.06 ± 0.77 nmol AFB$_1$ monoalcohol/mg protein/minute for females and males, respectively, p = 0.0039) and the turkey (1.75 ± 0.93 and 3.52 ± 1.55 nmol AFB$_1$

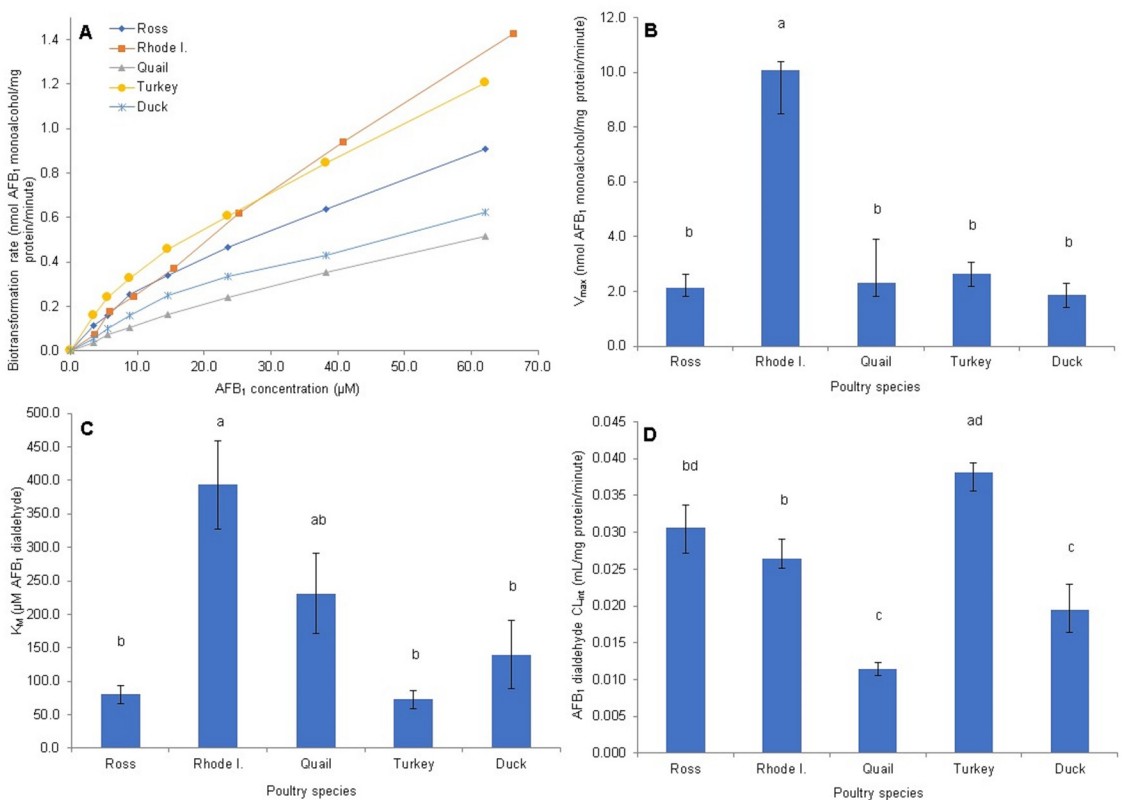

**Fig 4. Enzyme kinetic parameters of cytosolic *in vitro* AFB$_1$ monoalcohol production from AFB$_1$ dialdehyde.** (A) Saturation curve at AFB$_1$ dialdehyde concentrations of 3.4 to 66.4 $\mu$M (B) Maximal velocity (V$_{max}$). (C) Michaelis-Menten constant (K$_m$). (D) Intrinsic Clearance (CL$_{int}$; V$_{max}$/K$_m$). Species mean values sharing the same letter do not differ significantly. Statistical differences (P <0.05) were calculated using the Kruskal-Wallis test and nonparametric multiple comparisons were done by the Dwass-Steel-Critchlow-Fligner method. Values are means ± SEM of 12 birds.

monoalcohol/mg protein/minute for females and males, respectively; p = 0.025). The K$_m$ parameter value (Fig 4C) was significantly higher (p <0.0001) in Rhode Island Red chickens (393.5 ± 227.8 $\mu$M of AFB$_1$ dialdehyde) compared with the duck (139.4 ± 176.5 $\mu$M), the Ross chicken breed (80.2 ± 46.53 $\mu$M), and the turkey (72.7 ± 45.9 $\mu$M); however, it did not differ significantly from the quail (231.0 ± 206.1 $\mu$M). Differences in K$_m$ between sexes were found only for the Ross chickens, with values of 42.08 ± 23.8 and 118.4 ± 26.4 $\mu$M for females and males, respectively (p = 0.0039). The calculated intrinsic clearance (CL$_{int}$) for AFB$_1$-monoalcohol production was the highest in the turkey, and the Ross and Rhode Island Red chickens (0.038 ± 0.008, 0.031 ± 0.011, and 0.026 ± 0.004 mL/mg protein/minute, respectively). Significantly lower CL$_{int}$ values (p <0.0001) were observed for quail (0.011 ± 0.003 mL/mg protein/ minute) and duck (0.019 ± 0.010 mL/mg protein/minute; Fig 4D). Only the duck showed differences between sexes (0.02 ± 0.01 and 0.01 ± 0.004 mL/mg protein/minute for female and male, respectively; p = 0.0374).

AFB$_1$-dialcohol enzyme production kinetics is presented in Fig 5. In contrast to AFB$_1$-monoalcohol production activity, AFB$_1$-dialcohol production reached a plateau (enzyme saturation due to substrate concentration) below the highest AFB$_1$ dialdehyde concentration tested (66.4 $\mu$M); however, duck and quail reached the plateau at a lower substrate concentration compared with the two strains of chickens and the turkey (Fig 5A). A large difference in V$_{max}$ was found between the turkey, and the Ross and Rhode Island Red chickens (0.38 ± 0.17,

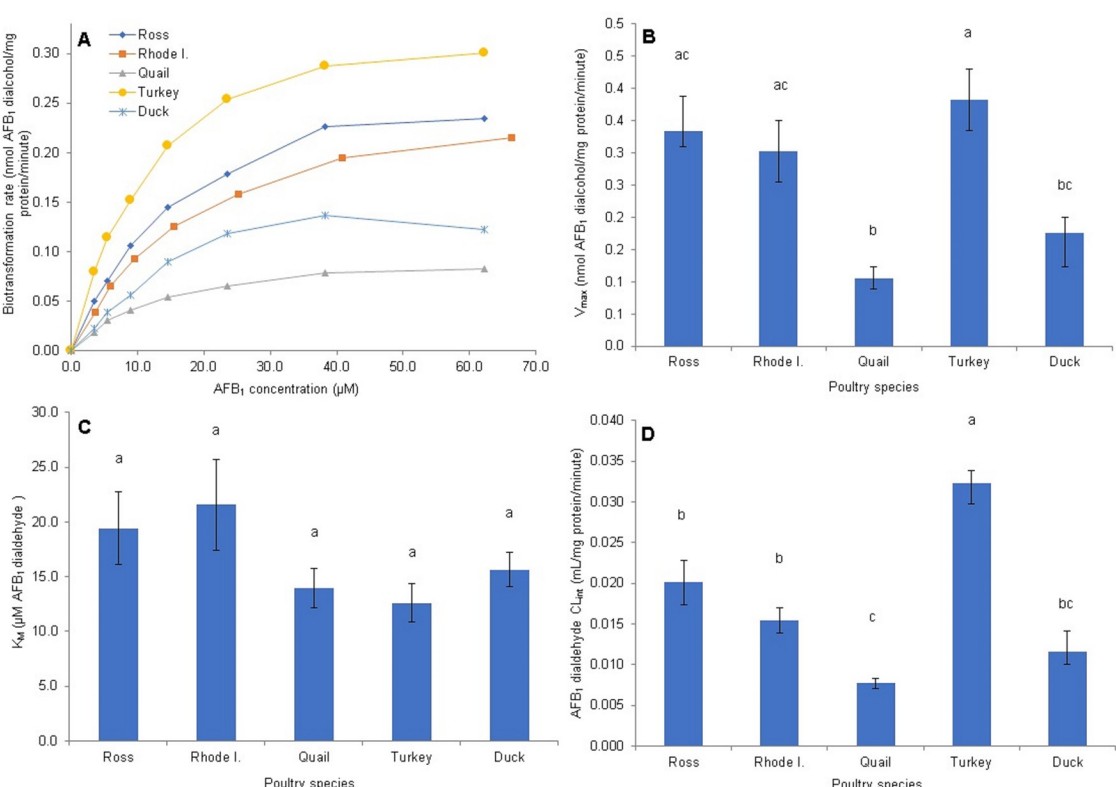

**Fig 5. Enzyme kinetic parameters of cytosolic *in vitro* AFB$_1$ dialcohol production from AFB$_1$ dialdehyde.** (A) Saturation curve at AFB$_1$ dialdehyde concentrations of 3.4 to 66.4 $\mu$M (B) Maximal velocity (V$_{max}$). (C) Michaelis-Menten constant (K$_m$). (D) Intrinsic Clearance (CL$_{int}$; V$_{max}$/K$_m$). Species mean values sharing the same letter do not differ significantly. Statistical differences (P <0.05) were calculated using the Kruskal-Wallis test and nonparametric multiple comparisons were done by the Dwass-Steel-Critchlow-Fligner method. No differences were found for K$_m$ enzyme activity parameter between poultry species. Values are means ± SEM of 12 birds.

0.33 ± 0.18 and 0.30 ± 0.17 nmol AFB$_1$ dialcohol/mg protein/minute, respectively) and the duck (0.18 ± 0.09 nmol AFB$_1$ dialcohol/mg protein/minute) and quail (0.11 ± 0.06 nmol AFB$_1$ dialcohol/mg protein/minute) (p <0.0001; Fig 5B). Differences by sex for this parameter were found in Rhode Island Red chickens (0.41 ± 0.17 and 0.20 ± 0.05 nmol AFB$_1$ dialcohol/mg protein/minute for females and males respectively, p = 0.0374), Ross chickens (0.21 ± 0.06 and 0.46 ± 0.18 nmol AFB$_1$ dialcohol/mg protein/minute for females and males respectively, p = 0.0104), and turkey (0.23 ± 0.03 and 0.53 ± 0.08 nmol AFB$_1$ dialcohol/mg protein/minute for females and males respectively, p = 0.0039). Even though there were numerical differences in K$_m$ values among the different poultry species, the differences failed to reach statistical significance (p = 0.4216). Values of K$_m$ were 21.57 ± 14.32 $\mu$M for Rhode Island Red chickens, 19.43 ± 11.62 $\mu$M for Ross chickens, 15.63 ± 5.47 $\mu$M for ducks, 13.97 ± 6.22 $\mu$M for quail, and 12.60 ± 6.12 $\mu$M for turkeys (Fig 5C). Only Ross chickens (10.62 ± 4.97 and 28.22 ± 9.33 $\mu$M for females and males, respectively, p = 0.0104) and turkey (7.20 ± 2.10 and 18.02 ± 2.77 $\mu$M for females and males, respectively, p = 0.0039) showed significant differences between sexes. Finally, the CL$_{int}$ value for AFB$_1$ dialcohol production (Fig 5D) was highest for the turkey (0.032 ± 0.008 mL/mg protein/minute) followed by Ross and Rhode Island Red chickens (0.020 ± 0.009 and 0.015 ± 0.005 mL/mg protein/minute, respectively), duck (0.012 ± 0.005 mL/mg protein/minute) and quail (0.008 ± 0.002 mL/mg protein/minute) with a p <0.0001. There were no significant differences between sexes.

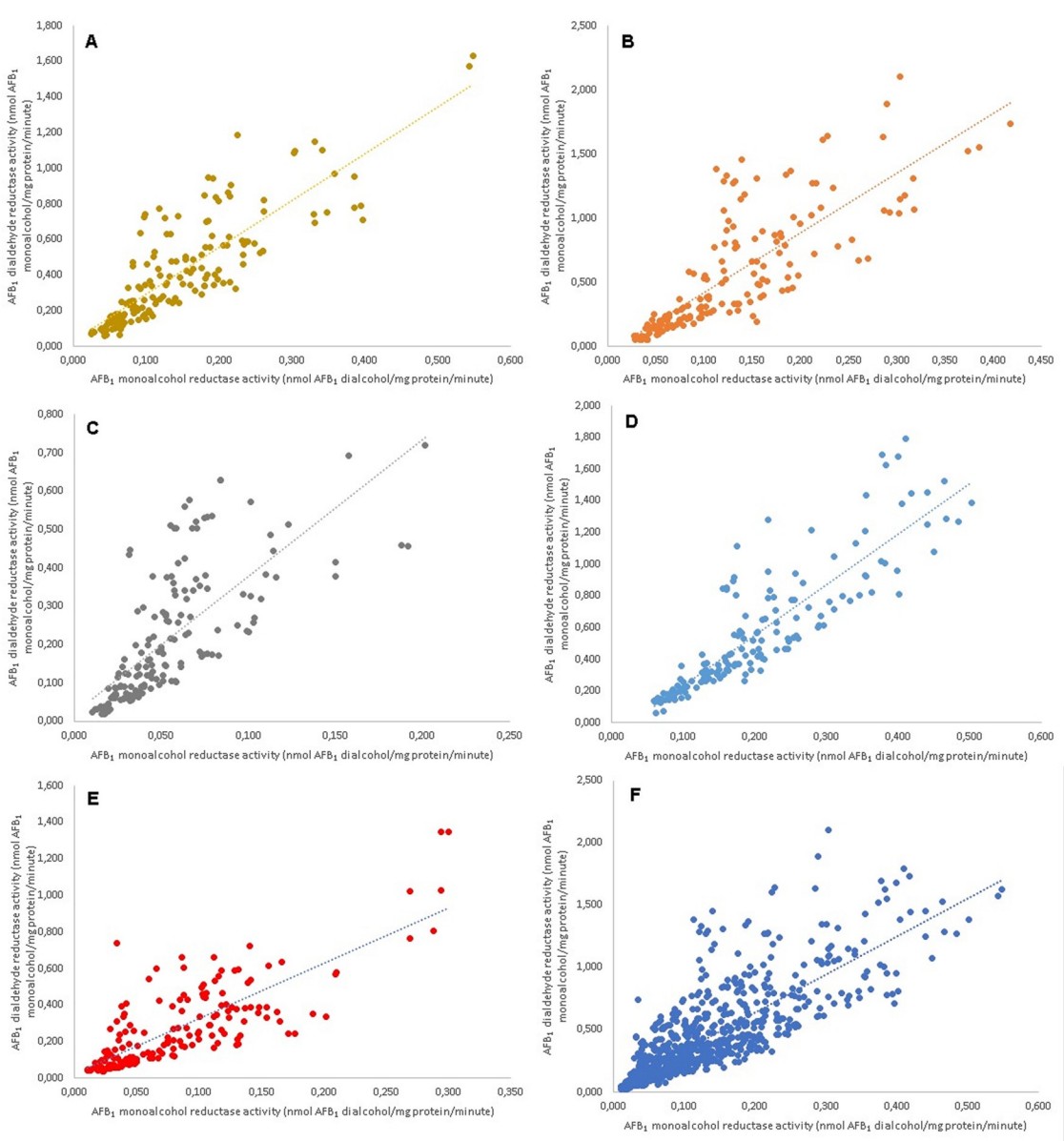

**Fig 6. Correlation of AFB₁ monoalcohol reductase enzyme activity vs AFB₁ dialdehyde reductase enzyme activity velocities.**
Cytosolic fractions from 12 birds were used and concentrations of AFB₁ dialdehyde from 3.6 to 66.4 $\mu$M. Spearman's rank-order correlation coefficient was calculated for A: Ross chicken breed (0.87), B: Rhode Island Red Chicken breed (0.88), C: quail (0.84), D: turkey (0.91), E: duck (0.78) and F: all species (0.86).

When the two enzymatic activities were compared, it was found that AFB₁-monoalcohol and AFB₁-dialdehyde reductase activities were significantly correlated ($p < 0.05$) for all poultry species. Spearman´s rank correlation coefficients for these two activities were 0.87 for Ross chickens, 0.88 for Rhode Island Red chickens, 0.84 for quail, 0.91 for turkey and 0.78 for duck. When the data obtained from all species were combined, the Spearman´s rank correlation coefficient value was 0.86 (Fig 6).

Finally, the ratios of CL$_{int}$ for AFB₁-monoalcohol production / CL$_{int}$ for AFB₁-dialcohol production and CL$_{int}$ for AFB₁-dialcohol production / CL$_{int}$ for AFB₁-monoalcohol

**Table 1. Comparison of CL$_{int}$ AFB$_1$-dhd enzyme production / CL$_{int}$ AFB$_1$ dialcohol enzyme production (dhd/dial), CL$_{int}$ AFB$_1$-dhd enzyme production / CL$_{int}$ AFB$_1$ monoalcohol enzyme production (dhd/mono), CL$_{int}$ AFB$_1$ monoalcohol enzyme production / CL$_{int}$ AFB$_1$ dialcohol enzyme production (mono/dial) and the inverse (dial/mono) ratios.** Among the different poultry species means were compared by using the Dwass-Steel-Critchlow-Fligner method. Values are means ± SD of 12 birds. Mean values with the same superscript do not differ significantly.

| Species | dhd/dial | dhd/mono | mono/dial | dial/mono |
|---|---|---|---|---|
| Ross chickens | 2.77 ± 1.84[a] | 1.65 ± 0.80[a] | 1.68 ± 0.68[a] | 0.67 ± 0.22[a] |
| Rhode Island Red chickens | 7.44 ± 2.37[b] | 4.01 ± 0.79[b] | 1.90 ± 0.65[a] | 0.59 ± 0.21[a] |
| Quail | 19.92 ± 11.89[c] | 12.99 ± 7.79[c] | 1.59 ± 0.63[a] | 0.70 ± 0.21[a] |
| Turkey | 3.89 ± 1.91[a] | 3.30 ± 1.55[b] | 1.22 ± 0.30[a] | 0.87 ± 0.23[a] |
| Duck | 167.80 ± 105.57[d] | 99.20 ± 62.08[d] | 1.92 ± 1.15[a] | 0.70 ± 0.39[a] |

production did not show significant differences among poultry species (p = 0.0846 and 0.0881, respectively; Table 1). However, when the ratios of CL$int$ for AFB$_1$-dhd production / CL$_{int}$ for AFB$_1$ monoalcohol production and CL$_{int}$ for AFB$_1$-dhd production / CL$_{int}$ for AFB$_1$ dialcohol production were calculated (based on AFB$_1$-dhd production data from the same set of samples [5]) significant differences were observed among the different species evaluated (p <0.0001; Table 1).

## Discussion

The *in vivo* sensitivity to AFB$_1$ in poultry species follows the order ducklings >>turkey poults >goslings >pheasant chicks >quail chicks >chicks [25]. Recent research conducted in our laboratory has shown that in poultry species the hepatic *in vitro* AFB$_1$-dhd production is related to the *in vivo* sensitivity, and we have hypothesized that AFB$_1$-dhd is the metabolite responsible for the acute toxic effects of AFB$_1$ [5]. AFB$_1$-dhd can exist in a pH-dependent equilibrium with AFB$_1$-dialdehyde (pKa value of 8.29) [4] and at a physiological pH of 7.2, AFB$_1$-dialdehyde can adduct lysine residues in proteins, leading to cytotoxicity. AFB$_1$-dialdehyde can be reduced to AFB$_1$-monoalcohol and AFB$_1$-diacohol by the AFAR enzyme. *In vitro* assays have shown that AFAR activity inhibits the formation of adducts with proteins [8], and therefore the alcohols can be considered detoxification products (there is no evidence that they can form adducts). Further, it has been observed that toxic dialdehydes like malondialdehyde (MDA) can be oxidized by mitochondrial aldehyde reductase, reducing its capacity to form adducts with nucleophile compounds like thiobarbituric acid [26]. In the present study, the V$_{max}$ for AFB$_1$ dialdehyde reductase enzyme activity (AFB$_1$-monoalcohol production) was highest for Rhode Island chickens (a resistant species), while the K$_m$ was the lowest for both resistant (Ross chickens) and susceptible species (turkey and duck). However, it is important to note that the K$_m$ parameter only reflects the enzyme-substrate complex dissociation constant [27]; in fact, the K$_m$ enzyme parameter is a collection of rate constants and not the binding constant for the interaction between enzyme and substrate, as it has been misunderstood [28]. Therefore, this parameter only indicates that Ross chickens, turkey and duck AFAR enzymes reach the V$_{max}$ at lower concentrations than Rhode Island Red chickens and quail. In the case of enzyme efficiency, measured as intrinsic clearance (CL$_{int}$), it was observed that highly resistant species (both chicken breeds) have the most efficient AFB$_1$-dialdehyde reductase enzyme activities. However, the turkey, which has an intermediate sensitivity between chickens and ducks, also had a high AFB$_1$-dialdehyde reductase activity; further, the quail, which is almost as resistant to AFB$_1$ as the chicken, had an AFB$_1$-dialdehyde reductase activity comparable to that of the duck. These results suggest that AFB$_1$-dhd detoxification by AFAR is related to poultry species resistance only in chickens and ducks. It is important to highlight

that AFAR enzyme activity cannot be considered as the only reaction capable of explaining the differences in sensitivity among different poultry species. For example, glutathione sulfotransferase (GST) enzyme activity is capable of affecting the production of AFB$_1$ dihydrodiol (and therefore the production of AFB$_1$ dialdehyde) through AFBO nucleophilic trapping. Therefore, the toxicity of AFB$_1$ should be considered as a multifactorial mechanism in which different metabolic pathways in AFB$_1$ biotransformation are interconnected, including AFAR activity. Regarding the AFB$_1$-monoalcohol reductase V$_{max}$ value, it was found that it does seem to be associated with species sensitivity since the chicken breeds had a higher value and the ducks a low value; however, the turkey is again an exception with a V$_{max}$ value similar to those found for the chicken breeds. Due to the fact that the K$_m$ value for this reaction did not differ significantly among poultry species, the AFB$_1$-monoalcohol reductase CL$_{int}$ values were dependent on the differences found in V$_{max}$. Therefore, the CL$_{int}$ for AFAR AFB$_1$-monoalcohol reductase showed differences between tolerant species (chicken breeds and quail) and the duck, but not for the turkey. Consideration must be given to the fact that AFAR expression and activity change with age [29] and this is probably the explanation for the higher AFAR activity found in the turkeys since they were older than 41-days (56 days-old), the age at which they are at the peak of AFAR activity.

The highly significant correlation found between AFB$_1$-dialdehyde and AFB$_1$-monoalcohol reductase activities strongly suggests that the same enzyme catalyzes both activities. This fact explains why the AFB$_1$-dialdehyde reductase activity did not saturate, even at the highest AFB$_1$-dialdehyde concentration used (66.4 $\mu$M of AFB$_1$-dialdehyde), whereas AFB$_1$-monoalcohol activity saturated completely. As the reduction of AFB$_1$-dialdehyde into AFB$_1$-monoalcohol moves forward, and the concentration of AFB$_1$-monoalcohol increases, the reduction of AFB$_1$-monoalcohol into the dialcohol saturates at lower AFB$_1$-dialdehyde concentrations, since both activities are being carried out by the same AFAR enzyme. This scenario is substantiated by the ratios of CL$_{int}$ for AFB$_1$-monoalcohol production / CL$_{int}$ for AFB$_1$-dialcohol production, which ranged from 1.22 to 1.92 (Table 1); these ratios indicate that AFB$_1$-monoalcohol production is favored over AFB$_1$-dialcohol production. Concerning the specific enzyme responsible for these reactions, it is most likely that the AFAR enzyme aldo-keto reductase AKR7A2 member is the one responsible for these two activities in the poultry species studied, according to the information provided by the NCBI [15] and the Kegg pathways [16–18] databases.

Recent findings have shown that AFB$_1$-dhd is probably the metabolite responsible for the acute toxic effects of AFB$_1$ since its hepatic *in vitro* production in related to the known *in vivo* sensitivity in poultry [5]. When the ratios of CL$_{int}$ for AFB$_1$-dhd production / CL$_{int}$ for AFB$_1$-monoalcohol production were compared it was found that AFB$_1$-dhd production is highly favored over AFB$_1$-monoalcohol production in the most sensitive species (the duck) compared with the other poultry species. The calculated ratios followed the order duck >>>quail >Rhode Island Red chickens = Turkey >Ross chickens. The ratios of CL$_{int}$ for AFB$_1$-dhd production / CL$_{int}$ for AFB$_1$ dialcohol production also followed the same pattern, suggesting that the cytotoxic effects of AFB$_1$ exposure in ducks are due to the lack of a detoxification pathway for the large amounts of AFB$_1$-dhd produced by their cytochrome P450 enzymes. A previous study failed to find an association between AFAR enzyme activity and animal resistance to AFB$_1$ exposure [30]. This apparent discrepancy could be the result of using the inappropriate model to determine V$_{max}$ and K$_m$ (the Lineweaver-Burk linearization method) since it is widely accepted that a nonlinear regression is a more accurate and precise method to estimate these parameters [31, 32].

## Conclusion

The present study provides, for the first-time, experimental evidence for the role of AFAR activity in the resistance to the acute toxic effects of AFB$_1$-dhd in different poultry species. AFB$_1$ dialdehyde and AFB$_1$ monoalcohol reductase enzyme activities (probably catalyzed by the AKR7A2 aldo-keto reductase family member) are higher in resistant species like the chickens, but also in less resistant like the turkey. Interestingly it was found that the ratio of CL$_{int}$ for AFB$_1$-dhd production / CL$_{int}$ for AFB$_1$-dialcohol production is more than a hundred times higher in the duck than in the chicken; this finding suggests that the duck is unable to cope with the highly unstable metabolite AFB$_1$-dhd, which results in acute toxic liver damage upon AFB$_1$ exposure. Finally, the correlation analysis between AFB$_1$-dialdehyde and AFB$_1$-monoalcohol reductase activities shows that some individuals posses high activity for both enzyme reactions; this fact suggests the possibility of selecting individuals with high rates of AFAR activity for the genetic selection of resistance, especially in sensitive species like the duck (S3 Table). The present trial is limited to the use of only one duck breed (Pekin breed), the number of individuals and the flock source of birds. Therefore, in order to identify possible tolerant individuals, a larger variety of birds should be assessed to validate more clearly the population effect across a wider diversity of bird sources. Additionally, possible interbreed differences should also be considered, since significant histopathological differences have been reported in different duck breeds after AFB$_1$ exposure [33].

## Supporting information

**S1 Table. Feed ingredients and nutritional content of the diets fed to the experimental birds.**
(DOCX)

**S2 Table. Final body weight and total feed intake at the time of sacrifice of the experimental birds.**
(DOCX)

**S3 Table. Values of CL$_{int}$ AFB$_1$-dhd enzyme production / CL$_{int}$ AFB$_1$ dialcohol enzyme production (dhd/dial), CL$_{int}$ AFB$_1$-dhd enzyme production / CL$_{int}$ AFB$_1$ monoalcohol enzyme production (dhd/mono), CL$int$ AFB$_1$ monoalcohol enzyme production / CL$_{int}$ AFB$_1$ dialcohol enzyme production (mono/dial) and the inverse (dial/mono) ratios per individuals.** Values in bold represent individuals which ratio value is below SD.
(DOCX)

## Acknowledgments

Diamir Ariza (Micotox Ltda., Bogotá D.C., Colombia) for the mass-spectrometric analysis.

## Author Contributions

**Conceptualization:** Hansen Murcia, Gonzalo J. Diaz.

**Formal analysis:** Hansen Murcia, Gonzalo J. Diaz.

**Funding acquisition:** Gonzalo J. Diaz.

**Investigation:** Hansen Murcia.

**Methodology:** Hansen Murcia.

**Project administration:** Gonzalo J. Diaz.

**Resources:** Gonzalo J. Diaz.

**Software:** Hansen Murcia.

**Supervision:** Hansen Murcia, Gonzalo J. Diaz.

**Visualization:** Hansen Murcia.

**Writing – original draft:** Hansen Murcia, Gonzalo J. Diaz.

**Writing – review & editing:** Hansen Murcia, Gonzalo J. Diaz.

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
