## [Decision Letter · Decision Letter 0]

19 Mar 2020

PONE-D-19-32966

Dealing with aflatoxin B1 dihydrodiol acute effects: impact of aflatoxin B1-aldehyde reductase enzyme activity in poultry species tolerance to AFB1 toxic effects

PLOS ONE

Dear Mr. Murcia,

Thank you for submitting your manuscript to PLOS ONE. After careful consideration, we feel that it has merit but does not fully meet PLOS ONE’s publication criteria as it currently stands. Therefore, we invite you to submit a revised version of the manuscript that addresses the points raised during the review process.

The manuscript should be revised deeply. The main problem found in the manuscript is related to the some aspects of methodology and poor discussion. Please review the referee comments and make your peer revision.

We would appreciate receiving your revised manuscript by May 03 2020 11:59PM. To enhance the reproducibility of your results, we recommend that if applicable you deposit your laboratory protocols in protocols.io, where a protocol can be assigned its own identifier (DOI) such that it can be cited independently in the future. For instructions see: http://journals.plos.org/plosone/s/submission-guidelines#loc-laboratory-protocols

We look forward to receiving your revised manuscript.

Kind regards,

Arda Yildirim, Ph.D.

Academic Editor

PLOS ONE

Additional Editor Comments (if provided):

The study is very well presented, I feel that the manuscript is dealing with a good topic but lacks in the quality of preparation. The main problem found in the manuscript is related to the methodology, other routes of detoxification were not discussed and typo errors or ambiguous phrases or sentences. It is necessary to improve the manuscript by examining the questions that need to be clarified in a way. Please be aware of the manuscript should be presented according to guidelines for authors of Plos One. For your guidance, you can check the reviewers' comments.

Journal Requirements:

2. We note that this submission reports a functional enzymological study with kinetic and thermodynamic data. The reporting of these data should include the temperature, pH and pressure, as well as the identity of the catalyst and its origins, the method of preparation, criteria for purity and assay conditions. We recommend that you refer to the Standards for Reporting Enzymology Data (STRENDA) of the Beilstein Institut for details regarding the adequate description of experimental conditions and reporting of enzyme activity data: https://www.beilstein-strenda-db.org/strenda/public/guidelines.xhtml. Please note that the Beilstein Institut’s STRENDA database automatically checks manuscript data for guideline compliance, as well as making them publicly available after publication and assigning them a specific DOI number for reference and tracking purposes. If you obtain a STRENDA Registry number (SRN) and PDF containing all your functional enzymology data, please include these as Supplementary files.

Reviewers' comments:

Reviewer's Responses to Questions

**Comments to the Author**

1. Is the manuscript technically sound, and do the data support the conclusions?

Reviewer #1: Yes

Reviewer #2: Yes

2. Has the statistical analysis been performed appropriately and rigorously? 

Reviewer #1: Yes

Reviewer #2: I Don't Know

3. Have the authors made all data underlying the findings in their manuscript fully available?

Reviewer #1: Yes

Reviewer #2: Yes

4. Is the manuscript presented in an intelligible fashion and written in standard English?

Reviewer #1: Yes

Reviewer #2: Yes

5. Review Comments to the Author

Reviewer #1: This manuscript provides an evidence that differences in the resistance of poultry species to aflatoxin B1 toxicity are associated with differences in AFAR activity.

Please consider the following suggestions:

Klein et al (Comp Biochem Physiol C Toxicol Pharmacol. 2002 Jun; 132 (2): 193-201.) show that AFAR acivity increases with age.

In this study, chickens were 7 weeks old, turkeys and quails were 8 weeks old, and Peking Duck was 9 weeks old. Please explain whether these age differences affect AFAR activity.

Different poultry species may have different feeds.

Please explain the potential effects of different feed types, compositions, and feeding patterns on bioenzymatic activity.

Ethoxyquin is an ingredient found in commercial feeds. It should be explained whether the diet used in this study contained ethoxyquin.

L307-309: The present study... in chickens and ducks.

...in different poultry species?

Reviewer #2: GENERAL COMMENTS

The MS provides a description of a study with potentially very important results. However, before the suggestions of selecting ducks with high AFAR activity, a larger variety of birds may need to be assessed to validate more clearly the population effect across a wider diversity of bird sources.

Is there any evidence already that some ducks may be dramatically less susceptible to AF exposure than others?

The limitation of the study should be included in the discussion.

It is not clear why other routes of detoxification were not discussed and this should be addressed. For example, enzyme activity in other organs may add to that of the activity in the liver (eg GST activity in the kidney).

SPECIFIC COMMENTS

TITLE

Change tolerance to tolerant: “… in poultry species TOLERANT to AFB1 toxic effects”

ABSTRACT

I suggest that the abstract would invite greater interest from potential readers if it stated more clearly WHAT IS KNOWN and WHAT IS ADDED BY THIS STUDY. It is an important and interesting subject, but one that is familiar to a very limited audience, an audience that could be increased by acknowledging that they are not abreast of the AFB1 literature.

As captured later, please consider and respond to the following points:

Intrinsic clearance: from a pharmacological perspective, clearance relates to a particular organ. In the biochemical context that is described in the MS it applies only to the selected enzyme. This should be made very clear, as hepatic clearance may be far higher than a single enzyme clearance if more than one enzyme is involved.

Highest AF dialdehyde concentration tested (66mcM): how does this concentration relate to the expected exposure concentration in intoxicated birds?

The same AFAR enzyme catalyses mono- and di-alcohol AFB1: is this the same enzyme or the same or related or linked enzyme?

AFAR activity related to toxic effects in chickens and ducks – not quail and turkey: does not this imply that the relationship is complex and potentially multifactorial in all species?

INTRODUCTION

No comments (all good)

MATERIALS AND METHODS

Lines 100-104

Details of the breed/source of the turkeys, quail, and Pekin ducks should be provided, especially to allow comparison with any further studies of in vitro or in vivo activity of AFB1.

In addition, the composition of the diet of the birds prior to sacrifice should be described, especially the presence of any medications in the diet. For example, it would be common practice to include anticoccidial agents in the diet of chickens and special anticoccidial agents can cause changes in hepatic enzyme activity. Similarly, the main nutritional ingredients in the diet should be identified.

Line 128

Is there a normal avian body temperature? Measurement of adult/mature bird body temperature has found a temperature of 41 for chickens, 41 for turkeys and 42 for ducks. Your comments on choice of a single temperature for all species and justification for the selection of 39 would be welcome. Younger birds may have a temperature closer to 39.

Lines 150-160

The statistical significance level should be presented and the actual value in each test presented later in the MS

Line 155

Intrinsic clearance – the units should be provided. In addition, as each species has a different mass of liver tissue, the total liver clearance for the enzyme of interest is a useful parameter.

RESULTS

Fig 2 and Fig 3 Captions

Why is the difference in MW of monoalcohol and dialcohol 2 units and not 1 unit?

DISCUSSION

Line 248

… its capacity to FORM adduct …

Line 264

The fact that there is an apparent relationship between AFAR activity and AFB1 susceptibility only in ducks and chickens, and not in quail and turkeys, suggests that toxicity may be multifactorial. Some discussion of this would add increased credibility to the study.

Line 273-274

Does the correlation described indicate that the same enzyme is involved or suggest that the enzyme activities are related or linked? Why is the only conclusion that they are the same enzyme?

Line 276

How representative of the concentration present in the liver of intoxicated birds is the highest concentration of AFBI-dialdehyde of 66mcM?

Line 298

Is the cytotoxic effect of AFB1 exposure in ducks DUE TO or ASSOCIATED WITH the lack of a detoxification pathway? What is the status of GST detoxification pathways in ducks?

CONCLUSION

Line 317

The mention of individuals with high activity may relate to the SD values in Table 1. It would be useful to provide the individual bird values from which the Table was compiled – in this way the variation by individual birds and the number of birds with significant variation can be seen – this could be included in a supplement.

Line 318-Line 319

Genetic selection would only be useful if there was a relationship between intoxication and enzyme activity. This would need to be established for the duck and may require challenge of ducks at the extremes (ie high and low) of enzyme activity with AF to see if in fact there is an in vitro in vivo (IVIV) relationship. Furthermore, only a very small sample of ducks from an unknown source has been studied and described herein.

6. PLOS authors have the option to publish the peer review history of their article (what does this mean?). If published, this will include your full peer review and any attached files.

Reviewer #1: Yes: Takeshi Kawasaki

Reviewer #2: Yes: Stephen Page

---

## [Author Response · Author response to Decision Letter 0]

15 Apr 2020

Reviewer #1: 

This manuscript provides an evidence that differences in the resistance of poultry species to aflatoxin B1 toxicity are associated with differences in AFAR activity.

Please consider the following suggestions:

Klein et al (Comp Biochem Physiol C Toxicol Pharmacol. 2002 Jun; 132 (2): 193-201.) show that AFAR activity increases with age.

In this study, chickens were 7 weeks old, turkeys and quails were 8 weeks old, and Peking Duck was 9 weeks old. Please explain whether these age differences affect AFAR activity.

Reply: We are in complete agreement with the reviewer about the changes on the expression levels of AFAR enzyme associated with the bird’s age, as it was reported by Klein et al. (2002). Unfortunately, the study of Klein et al. (2002) does not present western immunoblot densitometry values to compare between ages; however, their data suggest that AFAR expression reaches a maximum level in turkeys at 41 days of age. Further, other two key enzymes involved in xenobiotic biotransformation (cytochrome P450s and glutathione sulfotransferases) also seem to reach a maximum enzyme activity at 41 days of age in turkeys.

Since the turkeys used in our study were 8 weeks old (56 day-old), they were expected to have reached their maximum AFAR activity. To our knowledge there are no data regarding the relationship between age and AFAR activity in the other poultry species studied. A sentence has been inserted indicating the fact that AFAR expression and activity change with age and that consideration must be given to this fact when analyzing the results of the present trial (lines 297 - 300).

Different poultry species may have different feeds.

Please explain the potential effects of different feed types, compositions, and feeding patterns on bioenzymatic activity.

Reply: We agree with the reviewer in that different types of feed may affect the patterns of bioenzymatic activity. However, all diets were formulated with the same macro-ingredients (corn, extruded full-fat soybeans, soybean meal, vegetable oil, calcium phosphate, calcium carbonate, etc.), adjusted for the specific requirements of each poultry species. Therefore, we do not expect an effect of the diet on the results obtained.

Ethoxyquin is an ingredient found in commercial feeds. It should be explained whether the diet used in this study contained ethoxyquin.

Reply: No ethoxyquin was added to the diets provided to the experimental birds.

L307-309: The present study... in chickens and ducks. ...in different poultry species?

Reply: The words “in chickens and ducks” have been replaced by “in different poultry species” (lines 335 - 336). 

Reviewer #2: GENERAL COMMENTS

The MS provides a description of a study with potentially very important results. However, before the suggestions of selecting ducks with high AFAR activity, a larger variety of birds may need to be assessed to validate more clearly the population effect across a wider diversity of bird sources. Is there any evidence already that some ducks may be dramatically less susceptible to AF exposure than others?

The limitation of the study should be included in the discussion.

Reply: We agree with the reviewer in that the population of ducks used in our study is very limited. However, we chose the white Pekin duck in our study, which it is the type of duck most commonly used in commercial duck meat and egg production. We have added a sentence addressing this issue in the revised manuscript (lines 347 - 353).

It is not clear why other routes of detoxification were not discussed and this should be addressed. For example, enzyme activity in other organs may add to that of the activity in the liver (eg GST activity in the kidney).

Reply: We completely agree with the reviewer in that there are other routes of detoxication that are also important in AF metabolism. However, due to the fact that aflatoxin exposure is through the gastrointestinal tract and because all of the absorbed compounds from the GIT must pass first through the liver (because of the portal circulation), it is the liver the organ that is most important in AF biotransformation and detoxication. GST detoxication of the electrophilic metabolite of AFB1 (the 8,9-exo-epoxide, AFBO) is a key detoxication pathway that is also being investigated by our research group. What we want to point out in the present study is that tolerant poultry species like the chicken have a higher AFAR enzyme activity than sensitive poultry species like the duck. We do not intend to ignore the possible role of other metabolic pathways in AFB1 sensitivity. We only want to postulate that AFAR activity is probably contributing to the higher resistance of chickens to the adverse effects of AFB1.

SPECIFIC COMMENTS

TITLE

Change tolerance to tolerant: “… in poultry species TOLERANT to AFB1 toxic effects”

Reply: “Tolerance” has been replaced by “tolerant” in the title. 

ABSTRACT

I suggest that the abstract would invite greater interest from potential readers if it stated more clearly WHAT IS KNOWN and WHAT IS ADDED BY THIS STUDY. It is an important and interesting subject, but one that is familiar to a very limited audience, an audience that could be increased by acknowledging that they are not abreast of the AFB1 literature.

Reply: This is a very important suggestion and we would like to thank the reviewer for it. In order to create a greater interest in the article the following sentence has been added at the beginning of the abstract: “Aflatoxin B1 aldehyde reductase (AFAR) enzyme activity has been associated to a higher resistance to the aflatoxin B1 (AFB1) toxicity in ethoxyquin-fed rats. However, no studies about AFAR activity and its relationship with tolerance to AFB1 have been conducted in poultry. To determine the role of AFAR in poultry tolerance, the hepatic in vitro enzymatic activity of AFAR was investigated in liver cytosol from four commercial poultry species (chicken, quail, turkey and duck)”

As captured later, please consider and respond to the following points:

Intrinsic clearance: from a pharmacological perspective, clearance relates to a particular organ. In the biochemical context that is described in the MS it applies only to the selected enzyme. This should be made very clear, as hepatic clearance may be far higher than a single enzyme clearance if more than one enzyme is involved.

Reply: We are in complete agreement with the reviewer’s comments regarding CLint. 

We have inserted a sentence explaining that the calculated intrinsic clearance only applies for the selected enzymatic activity (not for the liver) and that due to the fact that we did not purify AFAR enzyme from hepatic cytosolic extracts, the calculated CLint values are only “apparent” [Apparent Intrinsic Clearance (CLintapp)] (lines 128 – 129, 167 – 170).

Highest AF dialdehyde concentration tested (66mcM): how does this concentration relate to the expected exposure concentration in intoxicated birds?

Reply: The AFB1 concentration expected to occur in the hepatocyte upon AFB1 exposure is in the nanomolar or even the femtomolar order (Ch’in, J. & Devlin, T. The distribution and intracellular translocation of aflatoxin B1 in isolated hepatocytes. Biochem. Biophys. Res. Commun. 122, 1–8). In chickens dosed with a single intra-crop bolus of 2 mg of AFB1 per kg of body weight (a very large amount compared to what it is expected from a contaminated diet) AFB1 plasma concentrations have been only 96 nM (Lauwers, M., Croubels, S., De Baere, S., Sevastiyanova, M. 2019. Supplementary materials: Assessment of dried blood spots for multi-mycotoxin biomarker analysis in pigs and broiler chickens. Toxins 11: S1 – S8). Therefore, it is likely that under natural conditions AFB1 dialdehyde concentrations are not going to reach the 66 µM concentration, because the cytosolic AFB1 concentration is expected to be in the nanomolar range. We used high AFB1 dialdehyde concentrations in our in vitro model because of the need to saturate the enzymes; also we needed to be able to detect the analytes (the limit of quantitation of the analytical technique is 0.4 ng/g for AFB1). 

The same AFAR enzyme catalyzes mono- and di-alcohol AFB1: is this the same enzyme or the same or related or linked enzyme?

Reply: The search conducted in databases like the NCBI or Kegg pathways indicates that the poultry AKR7A2 enzyme is responsible for the reduction of AFB1 dialdehyde and AFB1 monoalcohol. Because we found a strong relationship between these two enzyme activities (high Spearman’s rank-order correlation coefficient), we postulated that both activities are carried out by the same enzyme.

AFAR activity related to toxic effects in chickens and ducks – not quail and turkey: does not this imply that the relationship is complex and potentially multifactorial in all species?

Reply: We completely agree with the reviewer. AFAR activity is expected to depend on other reactions like glutathione sulfotransferase (GST) enzyme activity. In separate studies conducted by our research group we have found significant differences in GST activity among the same poultry species studied. Differences in the enzymatic nucleophilic trapping of AFBO will affect the production of AFB1 dihydrodiol and therefore the concentration of AFB1 dialdehyde. The overall metabolism of AFB1 most likely depends on several enzymatic reactions. A sentence has been added to clarify the multifactorial nature of the AFB1 detoxication pathways (lines 282 - 289).

INTRODUCTION

No comments (all good)

MATERIALS AND METHODS

Lines 100-104

Details of the breed/source of the turkeys, quail, and Pekin ducks should be provided, especially to allow comparison with any further studies of in vitro or in vivo activity of AFB1.

Reply: Details of the poultry used in the study have been provide as follows: Nicholas turkeys, Japanese quail (Coturnix coturnix japonica) (there are no Bobwhite quail -Colinus virginianus- in Colombia) and meat-type Pekin ducks (there are both meat-type and egg-type Pekin ducks (lines 102 - 104).

In addition, the composition of the diet of the birds prior to sacrifice should be described, especially the presence of any medications in the diet. For example, it would be common practice to include anticoccidial agents in the diet of chickens and special anticoccidial agents can cause changes in hepatic enzyme activity. Similarly, the main nutritional ingredients in the diet should be identified.

Reply: No additives or medication were added to the diets provided to the birds. The diets were formulated with the same ingredients (corn, extruded full-fat soybeans, soybean meal, vegetable oil, calcium phosphate, calcium carbonate, sodium chloride, lysine, methionine, tryptophan, choline, vitamin and mineral premix) formulated to reach or exceed the nutrient requirements of each poultry species studied. The ingredients and lack of use of non-nutritive additives have been indicated in the revised manuscript (lines 104 - 109).

Line 128

Is there a normal avian body temperature? Measurement of adult/mature bird body temperature has found a temperature of 41 for chickens, 41 for turkeys and 42 for ducks. Your comments on choice of a single temperature for all species and justification for the selection of 39 would be welcome. Younger birds may have a temperature closer to 39.

Reply: We agree with reviewer comment. Because the birds used in this trial were 7–9 weeks old birds, the expected temperature was around 39 degrees Celsius. The revised manuscript has been added with the words “the normal body temperature for the age of the birds used” (line 138 - 139).

Lines 150-160

The statistical significance level should be presented and the actual value in each test presented later in the MS

Reply: The following sentence has been added to the “Statistical analysis” section: “with a significance level of 5% (p<0.05).” (line 173). The actual p-value obtained in each test has been presented in the “Results” section where appropriate. 

Line 155

Intrinsic clearance – the units should be provided. In addition, as each species has a different mass of liver tissue, the total liver clearance for the enzyme of interest is a useful parameter.

Reply: The CLint units have been indicated: “mL/mg protein/minute” (lines 166-167). We agree with the reviewer comment about the concept of total liver clearance; however, since we did not purify the hepatic AFAR enzyme, we are not able to calculate the total liver clearance. We used the CLint value to express the enzyme efficiency in the same way that it has been reported in previous studies [Guengerich, F. P. Analysis and characterization of enzymes and nucleic acids relevant to toxicology. In Hayes’s Principles and Methods of Toxicology Sixth Edition (eds Hayes, A. W. & Kruger, C. L.) 1939, ISBN-13: 978-1842145364 (CRC Press, 2014)]. Further, as it was mentioned previously, the CLint value obtained corresponds to the apparent CLint.

RESULTS

Fig 2 and Fig 3 Captions

Why is the difference in MW of monoalcohol and dialcohol 2 units and not 1 unit?

Reply: AFB1 monoalcohol has an aldehyde function (R – CHO) that is reduced to alcohol, thus producing AFB1 dialcohol. In this carbonyl function, there is a double bond between carbon and oxygen (R – CH = O). When the carbonyl function is reduced, this C = O double bond becomes R – CH – O. One hydrogen is received by oxygen to produce the alcohol function (R – CH – OH) and the carbon atom receives the other hydrogen to complete its 4 bonds (R – CH2 – OH). That is why the molecular weight difference between the monoalcohol and the dialcohol is 2 units.

DISCUSSION

Line 248

… its capacity to FORM adduct …

Reply: The word “form” has been inserted as it was suggested (line 265)

Line 264

The fact that there is an apparent relationship between AFAR activity and AFB1 susceptibility only in ducks and chickens, and not in quail and turkeys, suggests that toxicity may be multifactorial. Some discussion of this would add increased credibility to the study.

Reply: We completely agree with the reviewer. The following paragraph has been added to the Discussion section: “It is important to highlight that AFAR enzyme activity cannot be considered as the only reaction capable of explaining the differences in sensitivity among different poultry species. For example, glutathione sulfotransferase (GST) enzyme activity is capable of affecting the production of AFB1 dihydrodiol (and therefore the production of AFB1 dialdehyde) through AFBO nucleophilic trapping. Therefore, the toxicity of AFB1 should be considered as a multifactorial mechanism in which different metabolic pathways in AFB1 biotransformation are interconnected, including AFAR activity” (lines 282 - 289).

Line 273-274

Does the correlation described indicate that the same enzyme is involved or suggest that the enzyme activities are related or linked? Why is the only conclusion that they are the same enzyme?

Reply: A linear correlation between two enzyme activities has been used in different experimental models to assume that a single enzyme is responsible for both enzyme activities (Eaton, D.L., Ramsdell, H.S., Neal, G. E. 1994. Biotransformation of aflatoxins. In: The toxicology of aflatoxins. Human health, veterinary, and agricultural significance. Eds. Eaton, D.L. and Groopman, J.D. Academic Press. 53 – 55; Shou, M., Dai, R., Cui, D., Korzekwa, K.R., Baillie, T.A., Rushmore, T.H. 2001. A kinetic model for the metabolic interaction of two substrates at the active site of cytochrome P450 3A4. J. Biol. Chem. 276: 2256 – 2262). Because of the high correlation between AFB1 dialdehyde reduction and AFB1 monoaldehyde reduction, we have hypothesized that the AKR7A2 member is likely responsible for both activities. 

Line 276

How representative of the concentration present in the liver of intoxicated birds is the highest concentration of AFBI-dialdehyde of 66mcM?

Reply: As it was explained previously, the 66 µM concentration of AFB1 dialdehyde was used due to the conditions required by the experimental model (although the expected values are in the nanomolar range, not the micromolar range). However, because reduction products are normalized to the amount of cytosolic protein used, the kinetic parameters obtained are expected to reflect what actually happens in the hepatocyte. 

Line 298

Is the cytotoxic effect of AFB1 exposure in ducks DUE TO or ASSOCIATED WITH the lack of a detoxification pathway? What is the status of GST detoxification pathways in ducks?

Reply: Currently there is no information on the enzyme kinetic parameters of GST activity against AFB1 or its metabolite AFBO in ducks. Kinetic parameters for this enzyme activity have been determined by our research group and these results are currently under consideration for publication. These unpublished studies have shown that the duck GST activity against AFBO is so low that the cytotoxic effects of AFB1 are most likely due to the lack of an appropriate detoxication pathway.

CONCLUSION

Line 317

The mention of individuals with high activity may relate to the SD values in Table 1. It would be useful to provide the individual bird values from which the Table was compiled – in this way the variation by individual birds and the number of birds with significant variation can be seen – this could be included in a supplement.

Reply: We agree with the reviewer regarding the identification of possible “high AFAR activity” individuals. Individuals with a dhd/dial or dhd/mono ratio value below average minus SD reflect individuals with the highest AFB1 dialdehyde reduction or AFB1 monoalcohol reduction efficiency (inactivation reactions) relative to the AFB1-dhd efficiency (activation reaction). The following chart has been added as supplementary material:

Species Sex Individual dhd/dial dhd/mono dial/mono mono/dial

ross female 1 2.73 2.72 1.00 1.00

ross female 2 1.09 0.74 0.67 1.48

ross female 3 1.74 1.57 0.90 1.11

ross female 4 1.40 0.64 0.46 2.20

ross female 5 2.88 1.29 0.45 2.23

ross female 6 3.99 2.31 0.58 1.73

ross male 1 1.27 0.94 0.74 1.36

ross male 2 1.21 1.16 0.96 1.04

ross male 3 7.27 2.13 0.29 3.41

ross male 4 4.73 2.70 0.57 1.75

ross male 5 1.69 1.04 0.62 1.62

ross male 6 3.25 2.66 0.82 1.22

island female 1 11.14 5.02 0.45 2.22

island female 2 11.13 3.49 0.31 3.19

island female 3 7.97 3.06 0.38 2.61

island female 4 5.74 4.04 0.70 1.42

island female 5 6.11 4.53 0.74 1.35

island female 6 6.16 3.73 0.61 1.65

island male 1 5.25 3.29 0.63 1.60

island male 2 3.53 3.84 1.09 0.92

island male 3 6.83 4.00 0.59 1.71

island male 4 10.08 4.23 0.42 2.38

island male 5 7.45 3.15 0.42 2.37

island male 6 7.93 5.76 0.73 1.38

quail female 1 18.22 18.44 1.01 0.99

quail female 2 6.73 6.22 0.92 1.08

quail female 3 9.67 9.28 0.96 1.04

quail female 4 8.18 6.94 0.85 1.18

quail female 5 29.19 17.06 0.58 1.71

quail female 6 6.34 3.76 0.59 1.68

quail male 1 40.84 29.43 0.72 1.39

quail male 2 17.77 12.99 0.73 1.37

quail male 3 14.70 7.25 0.49 2.03

quail male 4 34.83 10.65 0.31 3.27

quail male 5 19.59 9.87 0.50 1.98

quail male 6 33.03 24.02 0.73 1.38

turkey female 1 3.58 3.92 1.09 0.92

turkey female 2 9.48 7.64 0.81 1.24

turkey female 3 2.59 3.08 1.19 0.84

turkey female 4 3.63 2.85 0.78 1.27

turkey female 5 3.15 3.90 1.24 0.81

turkey female 6 3.05 3.38 1.11 0.90

turkey male 1 3.95 2.33 0.59 1.69

turkey male 2 3.63 2.36 0.65 1.54

turkey male 3 2.69 1.67 0.62 1.61

turkey male 4 2.83 2.31 0.82 1.22

turkey male 5 5.32 3.82 0.72 1.39

turkey male 6 2.81 2.35 0.84 1.20

duck female 1 208.75 81.43 0.39 2.56

duck female 2 228.83 59.49 0.26 3.85

duck female 3 361.73 84.35 0.23 4.29

duck female 4 177.11 85.12 0.48 2.08

duck female 5 44.56 23.13 0.52 1.93

duck female 6 101.50 82.99 0.82 1.22

duck male 1 181.93 182.60 1.00 1.00

duck male 2 60.95 87.24 1.43 0.70

duck male 3 102.01 139.39 1.37 0.73

duck male 4 58.63 38.46 0.66 1.52

duck male 5 142.59 81.06 0.57 1.76

duck male 6 345.03 245.13 0.71 1.41

S1 table. Values of CLint AFB1-dhd enzyme production / CLint AFB1 dialcohol enzyme production (dhd/dial), CLint AFB1-dhd enzyme production / CLint AFB1 monoalcohol enzyme production (dhd/mono), CLint AFB1 monoalcohol enzyme production / CLint AFB1 dialcohol enzyme production (mono/dial) and the inverse (dial/mono) ratios. Values in bold represent individuals which ratio value is below SD. 

Line 318-Line 319

Genetic selection would only be useful if there was a relationship between intoxication and enzyme activity. This would need to be established for the duck and may require challenge of ducks at the extremes (ie high and low) of enzyme activity with AF to see if in fact there is an in vitro in vivo (IVIV) relationship. Furthermore, only a very small sample of ducks from an unknown source has been studied and described herein.

Reply: We agree with the reviewer comment. We are fully aware of the limitations of the present study. Nonetheless we believe that we can postulate that research about genetic selection could potentially be used as a possible means of enhancing species tolerance. S1 Table gives additional support regarding the possibility of selecting individuals with a particularly high AFAR activity that could be selected for potentially higher tolerance to AFB1. As it has been mentioned in the first comment of reviewer #2, a sentence has been added to clarify the limitations of the present study in regard to the population studied (lines 347 - 353).

---

## [Decision Letter · Decision Letter 1]

19 May 2020

PONE-D-19-32966R1

Dealing with aflatoxin B1 dihydrodiol acute effects: impact of aflatoxin B1aldehyde reductase enzyme activity in poultry species tolerant to AFB1 toxic effects

PLOS ONE

Dear Mr. Murcia,

Thank you for submitting your manuscript to PLOS ONE. After careful consideration, we feel that it has merit but does not fully meet PLOS ONE’s publication criteria as it currently stands. Therefore, we invite you to submit a revised version of the manuscript that addresses the points raised during the review process.

Please review the referee comments on the term and giving supplemental data on the actual feed ingredients (mixing ratio and ME), feed intake, body weight, origin of rearing and notable clinical symptoms.

We would appreciate receiving your revised manuscript by Jul 03 2020 11:59PM. To enhance the reproducibility of your results, we recommend that if applicable you deposit your laboratory protocols in protocols.io, where a protocol can be assigned its own identifier (DOI) such that it can be cited independently in the future. For instructions see: http://journals.plos.org/plosone/s/submission-guidelines#loc-laboratory-protocols

We look forward to receiving your revised manuscript.

Kind regards,

Arda Yildirim, Ph.D.

Academic Editor

PLOS ONE

Additional Editor Comments (if provided):

Thank you for responding to all comments and for revising the manuscript, but there are still some flaws. Please review the referee#1 and 2 comments again, and make your final revision.

Reviewers' comments:

Reviewer's Responses to Questions

**Comments to the Author**

1. If the authors have adequately addressed your comments raised in a previous round of review and you feel that this manuscript is now acceptable for publication, you may indicate that here to bypass the “Comments to the Author” section, enter your conflict of interest statement in the “Confidential to Editor” section, and submit your "Accept" recommendation.

Reviewer #1: (No Response)

Reviewer #2: All comments have been addressed

2. Is the manuscript technically sound, and do the data support the conclusions?

Reviewer #1: Partly

Reviewer #2: Yes

3. Has the statistical analysis been performed appropriately and rigorously? 

Reviewer #1: Yes

Reviewer #2: Yes

4. Have the authors made all data underlying the findings in their manuscript fully available?

Reviewer #1: Yes

Reviewer #2: Yes

5. Is the manuscript presented in an intelligible fashion and written in standard English?

Reviewer #1: Yes

Reviewer #2: Yes

6. Review Comments to the Author

Reviewer #1: Thank you for responding to my comments and taking appropriate opinions.

However, in order to objectively understand the views expressed in this paper, there seems to be insufficient information on the experimental conditions as background.

To the author's opinion that dietary effects on the results obtained are not expected, it is necessary to be objective by presenting as much evidence as possible of each poultry rearing status in this study.

The authors should give supplemental data on the actual feed ingredients (mixing ratio and ME), feed intake, body weight, origin of rearing and notable clinical symptoms.

Reviewer #2: Thankyou for your detailed responses to the various questions raised.

The only matter you may wish to consider (and it is minor) is the use of DETOXICATION or DETOXIFICATION. The latter term is by far the most common term (a pubmed or a google search has around 10X as many entries for DETOXIFICATION compared with DETOXICATION).

7. PLOS authors have the option to publish the peer review history of their article (what does this mean?). If published, this will include your full peer review and any attached files.

Reviewer #1: Yes: Takeshi Kawasaki

Reviewer #2: Yes: Stephen W. Page

---

## [Author Response · Author response to Decision Letter 1]

2 Jun 2020

Review Comments to the Author

Reviewer #1: Thank you for responding to my comments and taking appropriate opinions. However, in order to objectively understand the views expressed in this paper, there seems to be insufficient information on the experimental conditions as background. To the author's opinion that dietary effects on the results obtained are not expected, it is necessary to be objective by presenting as much evidence as possible of each poultry rearing status in this study. The authors should give supplemental data on the actual feed ingredients (mixing ratio and ME), feed intake, body weight, origin of rearing and notable clinical symptoms.

Reply: We want to thank again reviewer #1 for his constructive comments. In order to clarify rearing origin and poultry clinical signs (the word symptoms is usually restricted to human patients), the following phrase has been added to manuscript: “Poultry were obtained from local commercial poultry suppliers and at the moment of sacrifice no noticeable clinical signs were observed “ (lines 109 - 111). 

In regard to the feed ingredient’s composition and calculated analysis (mixing ratio and ME), feed intake and body weight, tables S1 and S2 have been added as supplementary information. Former S1 table has been renumbered as S3 Table. 

Feed ingredients (%) Chicken diet Turkey diet Duck diet Quail diet

Corn 55.0 49.1 51.5 47.5

Corn gluten meal --- 4.5 5.0 3.5

Wheat bran --- --- 18.0 ---

Full-fat soybean (extruded) 11.7 6.0 2.0 6.8

Soybean meal (48%) 30.0 34.9 20.5 37.5

Vegetable oil 0.1 2.8 --- 1.0

Calcium carbonate 0.92 1.00 1.25 1.40

Calcium phosphate (20% P) 1.48 1.00 1.30 1.75

Iodized salt 0.3 0.2 0.2 0.35

Vitamin:mineral premix 1.0 1.0 1.0 1.0

Methionine 0.32 0.12 0.05 0.15

Lysine 0.10 0.25 0.12 ---

Threonine 0.03 --- --- ---

Calculated analysis (%)

Crude protein 25.1 25.5 20.1 26.1

ME (kcal/kg) 3103 3120 2800 2909

Ether extract 5.13 3.95 3.61 4.07

Crude fiber 2.77 3.24 4.08 3.33

Linoleic acid 1.25 1.49 1.42 1.29

 �-Linolenic acid 0.22 0.15 0.08 0.16

Calcium 0.95 0.88 0.99 1.24

Total phosphorus 0.62 0.54 0.66 0.69

Available phosphorus 0.30 0.25 0.33 0.34

Digestible lysine 0.74 1.41 0.96 1.50

Digestible methionine 1.36 0.48 0.38 0.49

Total sulphur amino acids 0.48 0.83 0.72 0.85

S1 Table. Feed ingredients and nutritional content of the diets fed to the experimental birds.

Species Sex Body weight (g) Feed Intake (kg/bird) Species Sex Body weight (g) Feed Intake (kg/bird)

Rhode Island Red chicks Female 750 1,07 Quail Female 89,1 0,84

 720 1,19 80,8 0,86

 625 0,99 79,2 0,86

 590 1,16 80,5 0,81

 630 1,18 83,0 0,85

 630 1,26 83,6 0,81

 Male 865 1,29 Male 92,9 0,81

 850 1,41 86,8 0,79

 830 1,37 96,1 0,82

 750 1,43 78,5 0,78

 810 1,27 73,6 0,81

 790 1,26 73,4 0,85

Ross chicks Female 1600 4,25 Turkey Female 2613 4,15

 2850 4,15 2674 4,25

 3175 4,23 3167 4,28

 2175 4,48 2600 4,26

 2450 4,15 2817 4,22

 3200 3,81 2627 4,26

 Male 3180 4,61 Male 3269 3,37

 2900 4,55 2912 3,28

 2750 4,78 2892 3,65

 2850 4,50 3036 3,56

 2550 4,78 2740 3,02

 2760 4,53 3197 3,44

Duck Female 1850 8,82 

 2640 8,69 

 2170 8,89 

 2300 8,92 

 2000 8,64 

 2100 8,70 

 Male 2700 7,77 

 2500 7,65 

 3100 7,79 

 2800 7,68 

 2600 7,95 

 2350 7,71 

S2 Table. Final body weight and total feed intake at the time of sacrifice of the experimental birds.

Reviewer #2: Thank you for your detailed responses to the various questions raised.

The only matter you may wish to consider (and it is minor) is the use of DETOXICATION or DETOXIFICATION. The latter term is by far the most common term (a pubmed or a google search has around 10X as many entries for DETOXIFICATION compared with DETOXICATION).

Reply: We want to also thank again reviewer #2 for his suggestion. The word “detoxication” has been replaced by “detoxification” were appropriate.

---

## [Editor Report · Decision Letter 2]

9 Jun 2020

Dealing with aflatoxin B1 dihydrodiol acute effects: impact of aflatoxin B1aldehyde reductase enzyme activity in poultry species tolerant to AFB1 toxic effects

PONE-D-19-32966R2

Dear Dr. Murcia,

We’re pleased to inform you that your manuscript has been judged scientifically suitable for publication and will be formally accepted for publication once it meets all outstanding technical requirements.

Kind regards,

Arda Yildirim, Ph.D.

Academic Editor

PLOS ONE

Additional Editor Comments (optional):

Thank you for responding to all comments and for revising the manuscript. Best regards,
---

## [Editor Report · Acceptance letter]

11 Jun 2020

PONE-D-19-32966R2 

Dealing with aflatoxin B_1_ dihydrodiol acute effects: impact of aflatoxin B_1_-aldehyde reductase enzyme activity in poultry species tolerant to AFB_1_ toxic effects 

Dear Dr. Murcia:

I'm pleased to inform you that your manuscript has been deemed suitable for publication in PLOS ONE. Congratulations! Your manuscript is now with our production department. 

Kind regards, 

on behalf of

Dr. Arda Yildirim 

Academic Editor

PLOS ONE